# Defect Creation in the Root of Single-Crystalline Turbine Blades Made of Ni-Based Superalloy

**DOI:** 10.3390/ma12060870

**Published:** 2019-03-15

**Authors:** Jacek Krawczyk, Robert Paszkowski, Włodzimierz Bogdanowicz, Aneta Hanc-Kuczkowska, Jan Sieniawski, Bartosz Terlecki

**Affiliations:** 1Institute of Materials Science, University of Silesia in Katowice, 1A 75 Pulku Piechoty St., 41-500 Chorzów, Poland; jacek.krawczyk@us.edu.pl (J.K.); wlodzimierz.bogdanowicz@us.edu.pl (W.B.); aneta.hanc@us.edu.pl (A.H.-K.); bterlecki@us.edu.pl (B.T.); 2Department of Materials Science, Rzeszow University of Technology, 2 Wincentego Pola St., 35-959 Rzeszów, Poland; jansien@prz.edu.pl

**Keywords:** superalloy, Bridgman technique, positron annihilation lifetime spectroscopy (PALS), vacancy-type defects, X-ray topography, defects, lateral growth, dendrite array, low-angle boundaries

## Abstract

An analysis of the defects in the vicinity of the selector–root connection plane occurring during the creation of single-crystalline turbine blades made of CMSX-6 Ni-based superalloy was performed. X-ray diffraction topography, scanning electron microscopy, and positron annihilation lifetime spectroscopy were used. Comparing the area of undisturbed axial growth of dendrites to the area of lateral growth concluded that the low-angle boundaries-like (LAB-like) defects were created in the root as a result of unsteady-state lateral growth of some secondary dendrite arms in layers of the root located directly at the selector–root connection plane. Additional macroscopic low-angle boundaries (LABs) with higher misorientation angles were created as a result of concave curvatures of liquidus isotherm in platform-like regions near selector–root connections. Two kinds of vacancy-type defects, mono-vacancies and vacancy clusters, were determined in relation to the LABs and LAB-like defects. Only mono-vacancies appeared in the areas of undisturbed axial growth. Reasons for the creation of macroscopic LABs and LAB-like defects, and their relationships with vacancy-type defects were discussed.

## 1. Introduction

It is known that many types of defects in single-crystalline (SX) turbine blades are created during directional crystallization by the Bridgman technique, while the crystallization front passes through areas with abrupt changes in a geometry of the cast [1,2,3,4,5,6,7,8]. The layers of the blade root located near the plane of connection with a selector (selector-root (S–R) connection), where branching of the primary dendrite arms and lateral growth occur, are especially important for defects creation [9,10,11,12,13,14]. In these layers, the low-angle boundaries (LABs), related to the local crystal misorientation, are created [15]. The LABs formed in the layers of the root, located near the plane of the S–R connection, may be inherited by the entire root and also by the airfoil [16,17]. The LABs are difficult or impossible to eliminate by heat treatment in subsequent stages of the blade’s production [18]. Furthermore, in some layers of the root located near the plane of the S–R connection, the dendrites grow in an unsteady regime, and additional defects such as dislocations and non-equilibrium vacancies may be created. Therefore, it is necessary to test for defects in the root layers located in the vicinity of the plane of the S–R connection and to analyze how they were created. The density of dislocations and vacancy-type defects are important for creep processes in Ni-based single-crystalline superalloys [19].

Studies of defects created by lateral growth may be performed by comparing defect structure areas created by undisturbed axial growth and the areas beyond it. Undisturbed dendrite growth occurs in the root area of the selector extension (SE, Figure 1a,b) as opposed to the neighboring areas (L, Figure 1b). The study of defects created during lateral dendrite growth are important, especially for the recently developed methods for obtaining single-crystalline casts, e.g., the liquid metal cast (LMC) method for SX cast production [20]. Disturbances in the expected crystallographic orientation may cause reduction in strain properties [21].

The defects in SX casts related to the local change of crystal orientation in macroscopic areas, e.g., macroscopic LABs or local change in the crystal interplanar spacing in these areas, may be visualized by X-ray reflective topography [22]. This method allows to indicate the macroscopic areas in which the changes in type and density of such defects as dislocations or vacancies may occur. All these changes may be specified by positron annihilation lifetime spectroscopy (PALS), which allows to obtain parameters of the macroscopic areas and is especially sensitive to vacancy-type defects. The common application of these two methods may allow to find a relation between the LABs and vacancy-type defects.

The PALS method, described in detail by Reference [23], is an exceptionally good method for the simultaneous determination of concentration and type of defects. The current PALS studies of Ni-based compounds are related to the characteristics of defects in Ni_3_Al-based alloys and in poly-crystalline Ni-based superalloys [24,25,26,27,28]. However, there are no PALS studies regarding investigations of defects in fragments of single-crystalline turbine blades.

The concentration of vacancy-type defects in multicomponent technical superalloys containing many alloying elements is particularly important because it is related to the kinetics of diffusion processes [29,30]. For example, it can determine the kinetics of the *γ* → *γ*′ solid-solution transformation during the Bridgman process or during heat treatment of SX turbine blades. The dendritic segregation of alloying elements is one of the reasons for the heterogeneity concentration of vacancy-type defects. The heterogeneity of vacancy-type defects concentration, especially the vacancy clusters, may be related to the pore distribution in as-cast blades [31]. Exceeding the critical concentration of the vacancies in the *γ* phase may cause creation of undesirable TCP phases [30,32]. On the other hand, the LABs, easily visualized by X-ray diffraction topography, are equally important because dislocations in LABs may play a role as sources or sinks for the vacancies. Additionally, in superalloys with the addition of W and Re (2nd and 3rd generations such as CMSX-4 and CMSX-10), the elements can also cause a specific phase instability related to the LABs’ presence [30,32].

The superalloys of the CMSX series are advanced materials commonly used in the aerospace industry for production of SX turbine blades [33]. Currently, the first generation of superalloys including CMSX-6 with a typical dendritic structure, created during directional crystallization, is the best examined. However, so far, the differences in the defect structure created by axial and lateral growth are unknown. This issue is particularly important for modern production methods such as LMC [20]. In the case of the conventional Bridgman process, it is possible to investigate a lateral growth by examining the layers of the root located near the S–R connection of the blades. The relationship between vacancy-type defects and LABs for CMSX-6 alloy has not yet been determined. Additionally, the mechanism for LABs creation in the layers of the root located near the S–R connection is poorly known. Therefore, the main goal of the presented paper is to examine the creation of defects and the relationships between them in the root layers located near the S–R connection, where lateral non-disturbed axial growth occurs.

X-ray diffraction topography is very useful to study the process of creating a dendrites array because it enables the visualization of such small misorientation of the neighboring regions and neighboring dendrites (angular minutes), that cannot be visualized by the other methods (e.g., EBSD). It allows to analyze the creation of defects during the lateral growth of separate dendrites and the creation of the dendrite array.

In the present work, a well-known Cannon–Muskegon 1st generation superalloy—CMSX-6—which does not contain such refractory elements as W and Re was selected for the study. The presented results for CMSX-6 refer to the first part of the study, and in the next part, the CMSX-4 and CMSX-4 Plus superalloys that contain W and Re will be studied, and for which the role of LABs is more significant compared to CMSX-6.

## 2. Materials and Methods

Blades with an axial orientation of the [001] type were obtained by the Bridgman technique using a spiral selector in an industrial ALD furnace at the Research and Development Laboratory for Aerospace Materials at the Rzeszów University of Technology. The withdrawal rate of 3 mm/min was used. The specimens for study were prepared from roots of CMSX-6 turbine blades. For the first step, a fragment T (Figure 1b,c) was cut-off from all blade roots in such a way that the plane *F* was parallel to the *Z* axis and crossed the fragment of the selector. Mechanical polishing and chemical etching were used for part of the *F** preparation of the *F* surface for SEM observations of the dendrite structure. Then, three slices (A, B, and C) were cut perpendicular to the withdrawal direction Z near the plane *M*–*N* (Figure 1a,b) of the S–R connection from the remaining root fragment H (Figure 1a). Three plate-shaped specimens, A, B, and C (Figure 1b), were prepared from the slices. The A and B specimens were used to form a sandwich layout for PALS measurements (Figure 1b).

The study procedures comprised the following steps: (1) SEM observations of the dendritic structure on the whole P_A_–P_A_ and P_B_–P_B_ surfaces (Figure 1b), (2) analysis of X-ray topograms received from the same surfaces, and (3) examination by PALS of three selected areas of each prepared sandwich. Area 1 of the A and B specimens was chosen because it was representative of the selector extension (SE) area (Figure 1a,b). It was assumed that in the SE area the axial growth of dendrites occurred directly from the selector without any structure disturbances. The other two areas were selected based on the beforehand obtained topograms: area 2 contained one or several macroscopic LABs, and area 3 did not contain such LABs. Additionally, the metallographic section was prepared from the surface *F** of the T fragment (Figure 1a) for the dendritic structure observation.

The dendritic structure was visualized by macro-SEM images (BSE technique) obtained by stitching several dozen SEM images together using a JEOL JSM-6480 microscope. For X-ray topography, the micro-focus X-ray CuK_α_ source provided by the Panalytical was used. The topograms were obtained using the 113 type reflections during coupled sample and film oscillation around the Bragg angle. Initially, the samples were crystallographically oriented by the Laue method.

The PALS measurements were performed at room temperature in a sandwich geometry using a “fast–fast” coincidence spectrometer. The positron source with an activity of about 370 kBq covered with a 5-μm Ni foil was used. The data were analyzed using the LT software [34] and were further fitted using the least squares fitting procedure to a simple three-state-trapping model implemented directly to the LT code [34]. This model describes a positron lifetime in the bulk material *τ_b_*, two lifetimes of positrons trapped inside two different types of defects *τ*_1_ and *τ*_2_, and two respective positron trapping rates *κ*_1_ and *κ*_2_. The values of *κ*_1_ and *κ*_2_ are proportional to the atomic concentrations of both types of defects. The positrons are trapped in vacancy-type defects (Figure 1d), where the electron density is reduced. It results in a longer positron lifetime in comparison to the bulk. The positron annihilation in the bulk and the defects cause the emission of *γ* radiation. Measured time intervals between implementation and annihilation of positrons allows to determine their lifetime and the concentration of vacancy-type defects.

## 3. Results and Discussions

In the lateral macro-SEM images of the H-type samples obtained from P_A_–P_A_ and P_B_–P_B_ surfaces, two systems of perpendicular secondary dendrite arms parallel to the *p* and *q* directions were found (Figure 2a). In the D_1_ and D_2_ areas, marked by white dotted lines, the dendritic structure was finer than in the other part of the sample. In area D_1_, the dendrite arms were arranged in long straight chains (SCs) parallel to *q*. A similar distribution of the secondary dendrite arms occurred in the D_2_ area, but the SCs were parallel to *p* and *q*. Some of the longest SCs were indicated as L_1_, L_2_ (parallel to *q*), and L_3_ (parallel to *p*) in Figure 2a. In the middle part of the specimen, corresponding to the SE area (circular area, Figure 2a), the SCs occurred much less frequently. The SCs were also present around the SE area.

Figure 2b presents an example of an X-ray topogram obtained for the P_A_–P_A_ surface. The topogram of the P_B_–P_B_ surface was similar. Three sub-grains were visible: SG_A_, SG_B_, and SG_C_ (Figure 2c,b), separated by wide contrast bands of BI and BII with decreased contrast. The bands corresponded to the macroscopic LABs, LAB-I and LAB-II, respectively. Such LABs are characteristic for Ni-based superalloys [9,16,18]. Additionally, in the topogram, groups of long bands of increased and decreased contrast were visible. They are marked by L_1_′–L_3_′ in Figure 2b,d and by L_4_′–L_7_′ in Figure 2d (L_4_′–L_7_′ were not marked in Figure 2b to increase the clarity of the topogram). Moreover, the circular area without long bands in the region corresponding to the SE area may be observed. The bands on the topograms were parallel to the *p*′ and *q*′ directions, which were parallel to [010] and [100] directions defined by the Laue diffraction. These bands corresponded to the *p* and *q* directions of the secondary dendrite arms (Figure 2a), although the *p*′ and *q*′ directions were not perpendicular to each other in the topogram. Additionally, the contour of the topogram was slightly different than the outline of the macro-SEM image, consistent with the shape of the sample. The difference results from the fact that the diffraction planes of the (113) type used for topograms were not parallel to the surface tested by SEM. The angle between the surface of the samples and the planes of the (113) type was *β* = 20° (Figure 2e). Neighboring bands of low- and high-contrast intensity, such as S_1_ and S_2_ (Figure 2d), were created as a result of the bands’ shift in the topograms (Figure 2e). Such a shift indicates a crystal misorientation of neighboring sample areas, and bands with this shift are interpreted as a macroscopic LAB-like, because they have a different character than LAB-I and LAB-II. When comparing such bands (L′_1_–L′_3_ in Figure 2b or L′_4_–L′_7_ in Figure 2d) with the macro-SEM image it can be observed that they were formed in the SCs regions of the dendritic structure. It means that when the secondary arms form the SC it causes the misorientation of the entire chain in relation to the neighboring part of the dendritic structure. In the SE part, this effect does not occur or is much weaker.

It should be emphasized that on the basis of the macro-SEM images only, it is difficult or not possible to identify the LABs or LAB-like type and SE type area. However, the location and shape of possible disturbances in the dendritic structure may be indicated in the topograms [18].

Based on the topogram presented in Figure 2b, the components *φ_F_* and *φ_H_* of misorientation angles for the LAB-I and LAB-like L_1_ were calculated using the Formulas (1) and (2) [35]: (1)$$\sin\varphi_{F}=\frac{1}{2\sin\theta_{113}}\cdot \frac{S_{H}}{\sqrt{S_{H}^{2}+R^{2}}}$$
(2)$$\sin\frac{\varphi_{H}}{2}=\frac{S_{F}}{2R}\cdot \sin\left(\theta_{113}-\beta \right),$$
where *S_H_* and *S_F_* are the shifts along the H and F axes measured in the topogram, *θ*_113_ is the Bragg angle equal to 47°, *β* is the angle between the surface of the samples and diffraction planes equal to 20° (Figure 2e), and *R* is the sample film distance equal to 78 mm. The total misorientation angle for LAB-I and LAB-like L_1_ was calculated based on Formula (3):(3)$$\varphi =\sqrt{\varphi_{H}^{2}+\varphi_{F}^{2}+\varphi_{P}^{2}}$$
where component *φ_P_* was calculated based on the additional topograms obtained from the surfaces S_1_ and S_2_ (Figure 2a). The calculated values of *φ_I_* for LAB-I and *φ_L_*_1_ for LAB-like L_1_ were 1.05° and 0.64°. The misorientation value of LAB-like L_1_ will be further marked as *φ_L_*, which is characteristic for LAB-like.

The PALS analysis was performed in areas 1, 2, and 3 (Figure 2d) with the use of positron lifetime *τ_b_* in bulk material as a common parameter. The determined value of *τ_b_* was 130 ± 1 ps. This value was a bit higher than the theoretical (110 ps [36,37]) and experimental values for Ni_3_Al (112 ps [38] and 125 ps [39]). The differences were related to large amounts of CMSX-6 alloying elements, e.g., Cr, Co, Mo, Ti, Ta, and Hf [33]. The fitting of the simple tree-state trapping model allowed to determine the positron trapping rates *κ*_1_ and *κ*_2_ for each area from 1 to 3. The values of *κ* were used for estimation of vacancy concentration (*c_V_*) based on the equation *c_V_* = *κ*/*µ*. In these calculations, it was assumed that the positron trapping coefficient (*µ*) did not depend on the local environment of vacancy and was 2.5 × 10^14^ s^–1^ [38]. The essential results of the PALS analysis are shown in Figure 3. Based on the theoretical results and obtained values, it was found that *τ*_1_ and *τ*_2_ corresponded to annihilation lifetime in mono-vacancies and in vacancy clusters, respectively, with the nickel vacancy behavior.

The obtained values of positron lifetimes (178 ps) indicated that in area 1 (Figure 3), only mono-vacancies in the *γ*′ phase were present. It can be assumed that the high structure homogeneity was the result of undisturbed axial growth of the primary dendrites from the selector toward the inside of the SE area. In areas 2 and 3, two types of defects were identified. Apart from the mono-vacancies in the *γ*′ phase, vacancy clusters were also found. The area containing LABs of type I (area 2) had a higher mono-vacancies concentration (Figure 3, marked as hatched bars) in comparison to the areas without LABs similar to LAB-I (area 3).

The PALS results show that the concentration of the vacancy clusters (Figure 3, marked as cross-hatched bars) was also higher in area 2 compared to area 3. However, the values of the positron lifetime in area 3 (*τ*_1_ = 185 ps, *τ*_2_ = 197 ps) were similar as in area 2 (*τ*_1_ = 183 ps, *τ*_2_ = 209 ps), which were related to the presence of mono-vacancies and vacancy clusters in both areas. The slight different positron lifetime values were probably related to the differences in the local environment of vacancy-type defects. The differences may be also related to the different local crystal misorientation in areas 2 and 3. Additionally, for area 2, it can also be related to the presence of LAB-I-type (Figure 2b–d), composed of a net of dislocations with increased concentration. It may cause an increase in the vacancy concentration, because *φ_I_* > *φ_L_* suggests a higher dislocation density.

The positron lifetime that corresponded to dislocations was not found for any of the three areas. It confirms the assumption that the annihilation rate for the vacancy-type defects was higher than for the dislocations, and all positrons emitted from the source were quickly trapped and annihilated on vacancy-type defects but not on dislocations. The mono-vacancy concentration determined by the PALS method applied to both equilibrium vacancies and non-equilibrium vacancies. Non-equilibrium vacancies may be related to the heterogeneity of chemical composition and occurrence of pores or dislocations [31]. The differences in the concentration of mono-vacancies in areas 1, 2, and 3 were related to the change of non-equilibrium defects only, assuming that in the macroscopic areas of 1, 2, and 3, where the positron source was active, the average macroscopic chemical composition and temperature were the same. The clusters of vacancy were non-equilibrium defects, so the values and changes of the concentration in areas 2 and 3 were directly defined by PALS.

The topogram presented in Figure 2b was formed by contrast related to the crystal orientation. Therefore, different details in the topograms were related to different distributions and values of local crystal orientation. Based on the generally accepted assumptions that all dendrite arms grow exactly in the <001> type direction, it may be concluded that topograms visualize the local change of the dendrites’ growth directions. Analysis of the topograms shows that the bands of misoriented areas type L′_1_–L′_7_, (Figure 2b,d) occurred outside the SE area, e.g., in areas 2 and 3 of the PALS measurement. The areas of the PALS sandwich (Figure 1b) were located in specimens A and B, placed above specimen C, inside which the unsteady lateral growth of the secondary dendrite arms in directions L (Figure 1b) may appear. The lateral growth occurs in the layer located directly above plane of S–R connection and may affect the creation of the dendrite array in the specimens A and B. Therefore, the microstructure of specimen section *F** (Figure 1a) were studied in detail.

Figure 4a shows the dendritic structure visualized on the surface *F** of the *F* section. Figure 4b shows the scheme of the array of dendrite arms that were roughly parallel to the axis Y. The scheme was prepared by creation of skeletal images and modification of contrast. Skeletonization is a technique whereby a binary image of dendrites is eroded away step by step, until the skeleton of the image is obtained. The skeletal image is created as a thin line equidistant from the original edges of the binary dendrites’ shape [18].

The surface *F** of the *F* section (Figure 4b) may be divided into four areas corresponding to different morphologies of the dendrite array: I, II, III, and IV, and which may be named as growth areas. Additionally, the *q* and *g* lines were marked. The line *q* crossed the area I and IV at the level of Y_q_. The line *g* crossed the area I and II at the level Y_g_. The area I was a fragment of SE (Figure 4b), where the primary arms grew directly from the selector without structural disturbances. The distance between the primary dendrite arms visualized in area I is denoted as the linear distance of the primary arms (LDPA_I_) unlike the primary arm spacing (PAS), determined usually in the transverse section [20]. The dendritic growth roughly along the *y*-axis in areas II, III, and IV was followed by tertiary arms, growing from some secondary arms of area II, marked by arrows in the left side of Figure 4a,b. The distance between tertiary dendrite arms in areas II, III, and IV is denoted as the linear distance of tertiary arms (LDTA_II_, LDTA_III_, and LDTA_IV_, respectively). These distances and LDPA_I_ were measured between the axis of the neighboring arms along the line perpendicular to the primary and tertiary dendrite arms. The LDPA_I_ and LDTA_IV_ refer to the primary and tertiary dendrite arm spacing (PAS and TAS). The PAS and TAS values were determined by classical method with standard deviation in the microstructure of the transverse section of the T fragment of the roots (Figure 1a). It was measured that PAS = 502 ± 92 µm and TAS = 355 ± 70 µm. The RS segment in Figure 4b corresponds to the segment RS at the bottom root surface (Figure 1a,c). In growth area I, the LDPA_I_ values were similar and amount to an average of 515 ± 90 μm. Growth area II, with a thickness that fluctuated from 0.5 to 2 mm, was located between X = 0 and X = X_N_ (Figure 4b). The LDTA_II_ value in the area II was constant along the *y*-axis and gradually increased along the *x*-axis. In Figure 4b, above growth area II, there was growth area III with high changes in LDTA_III_. In the left fragment of growth area III, the LDTA_III_ value increased stochastically in the direction of the *y*-axis in such a way that near Y = Y_0_, the LDTA_III_ became constant and equal to LDTA_IV_. In the right fragment of growth area III, the LDTA_III_ was approximately constant and equal to LDPA_IV_. In the fragment of line *q* (from X = 0 to X = X_N_), the LDTA_IV_ was close to the LDPA_I_ and amounted to 490 ± 90 μm. This means that the average linear distance of primary (growth area I) and tertiary (growth area IV) dendrite arms were similar from X = 0 to X = X_S_ in the *q* line (Figure 4b).

The relationship between the linear distance of the primary and tertiary arms denoted, as LDA and the X values along the *g* and *q* lines marked as LDA_g_(x) and LDA_q_(x), as well as hypothetical shape of the crystallization fronts, are presented in Figure 5. Lines visible in Figure 5 are the trend lines.

It can be seen that the LDA_q_(x) was approximately constant and the LDA_g_(x) increased from x = 0 to X = X_N_, reaching the value of the LDA_q_ in the vicinity of X_N_. The above dendritic array creation and character of the LDA_g_(x) and LDA_q_(x) relations can be described as follows. Growth area II was a layer in which lateral growth occurred under non-steady-state conditions. A selected secondary dendrite arm (arrow in Figure 4a,b) inclined at a small angle α of about 6° in relation to the *x*-axis, grew rapidly from the SE area. Fine tertiary dendrite arms grew from this arm approximately along *y*-axis. The growth rate of the secondary arms and tertiary arms in growth area II was much higher than the growth rate of the primary arms in growth area I, which was revealed by a much lower LDTA value about 100 μm (Figure 5a). In transition growth area III, some of the tertiary dendrite arms ceased to grow as a result of competitive growth process, and the LDTA increased to reach the value of the LDTA_IV_ in growth area IV. In this area, the dendrites’ growth took place already in steady-state conditions, corresponding to the withdrawal rate of 3 mm/min, as in growth area I (SE) of undisturbed growth. The secondary dendrite arm marked in Figure 4a by the arrow may be named the leading arm. These type of dendrite arms create the chains of tertiary arms, which after steadying conditions, form the SCs coming from the SE (e.g., L_3_), visible in the transverse microsection and topogram (Figure 2a,b). If it is assumed that the crystallization front moves in the direction Y at 3 mm/min, the rate of axial growth is *V_ax_* ≈ 3 mm/min. (Figure 6). With the additional assumption that the crystallization front is perpendicular to *V_ax_* and is flat from X = 0 to X = X_S_, the lateral growth rate *V_L_* of the leading secondary dendrite arm can be related to *V_ax_* and expressed by the formula:(4)$$V_{Le}=V_{ax}/\sin\alpha ,$$
where *α* is an inclination angle of the leading arm. For example, if *α* = 6° then sin 6° ≈ 0.1, therefore, the *V_L_* is one order of magnitude higher than *V_ax_*. However, when the distance from the SE is higher (i.e., at X → 0), the fragments of the crystallization front do not keep up with the movement of the front towards *V**_ax_* (Figure 5b) in the SE area, and a convex bending of the solidification front is created (front 1, Figure 5b). The local delay of the crystallization front fragments is described by the ΔY value in Figure 5b. The value of ΔY increases when approaching X = 0. If ΔY increases, the local undercooling ΔT must increase and the LDTA decrease (Figure 5a). A similar mechanism is presented in Reference [20]. On the other hand, the lateral heat dissipation can cause concave bending of the solidification front (front 2, Figure 5b) [7,20,40,41]. There is an accumulation of the abovementioned effects for growth area II resulting in formation of real crystallization front shape.

The L′_1_–L′_7_ bands in the topogram are parallel to the SCs, while LAB-I and LAB-II have another arrangement. They are located near the side walls of the root and are almost parallel to these walls. These types of LABs may be a result of the concave curvature of isotherm [6,7], solid–liquid interface, and crystallization front.

The rate of lateral growth in growth area II in unsteady-state conditions was much higher than axial growth of the primary arms in the SE region. This is the reason for the crystal misorientation of the dendrites in SCs originating from the leading secondary dendrite arm in growth area II. These misorientation defects were inherited by the areas of the root which crystallized later in steady-state conditions. This inheritance takes place through the growth of the tertiary dendrite arms after finishing the competitive growth process in transition area III. Therefore, it can be concluded that the formation of SCs are the reason for the LAB-like creations. Inside the SE region, the SCs did not occur or occurred rarely.

The differences in the values of the misorientation angle of LABs and LAB-like defects, as well as the differences in array and morphology of the dendrites are related to changes in concentration and type of vacancy defects. On the one hand, only mono-vacancies were present in the SE areas, and on the other, the mono-vacancies with the vacancy clusters were present in the areas with bands of crystal misorientation, visible as the SCs (area 3, Figure 2c). This means that the vacancy clusters were related to the crystal misorientation of the secondary arms in the SCs. In area 2 of the PALS measurements, the concentration of vacancy clusters and mono-vacancies was higher when compared to area 3. There were misoriented contrast bands in area 3 originating only from the SCs, similar to area 2, where the LAB-I, with a much higher misorientation angle, was additionally present.

The formation of vacancy clusters in areas containing low-angle boundaries (e.g., area 2 of PALS measurement, Figure 2d) was a complicated process that may be based on two basic phenomena: diffusion of vacancies to dislocations and the Kirkendall effect in *γ* phase, mainly at temperatures above the solvus, as what was described for nickel-based superalloys in Reference [42]. The low-angle boundaries created by locked-dislocation systems cause diffusion of vacancies toward them (Figure 7a). As a result, the mono-vacancy accumulation occurs at the LABs. It should be noted that during solidification, the alloying elements are segregated in the diffusion area of liquids, in such a way that the majority of the high-melting elements of Co, Mo, and Cr diffuse into dendrites, and the low-melting elements of Al and Ti into the inter-dendritic regions. Between these regions of the *γ* phase (above the solvus temperature), a gradient of the concentration of elements was formed. In the further crystallization step, the *γ* → *γ*′ transition takes place in the solid phase. The concentration gradient causes the high melting elements Co, Mo, and Cr to diffuse from the dendrite into the inter-dendritic region, and the elements with a lower melting point, such as Al and Ti, go in the opposite direction. Since there are different diffusion coefficients for both types of elements, according to Kirkendall’s theory, the vacancies were created by a diffusion mechanism. In the inter-dendritic region, additional vacancies appeared (light elements have a higher diffusion coefficient and leave their positions faster than they are planted by Co, Cr, and Mo). Created vacancies diffuse also to low-angle boundaries, where they accumulate, and their concentration becomes so high that it is sufficient to form clusters. Moreover, because vacancy clusters appear mainly through heterogeneous nucleation, a probability of the concentration in LABs was higher [43].

The creation of vacancy clusters in superalloys with a lot of alloying elements can be done through other, more complex mechanisms, related to the alloying elements’ interactions and preferences in occupancy of the lattice node. In area 1 of the PALS measurements (Figure 2d), vacancy clusters were not created because there was no LAB and LAB-like defects, and the concentration of vacancies was not yet reached, allowing for the creation of clusters. In area 2 (Figure 2d), the density of vacancies and clusters was higher than in area 3, because the LAB in area 2 had a larger misorientation angle and higher dislocation density.

The vacancies of the *γ*′ phase identified by PALS in areas 2 and 3 (Figure 2d) were formed mainly in the *γ* phase at high temperatures above the *γ* → *γ*′ transition. Let us consider the process of vacancies inheritance from *γ* to *γ*′ phase. The processes of diffusion of vacancies to dislocations and the Kirkendall effect described above, take place mainly in the *γ* phase below the solidus and above the solvus temperature. Creation of *γ/γ’* structures are described for dendrites by paths marked on fragments of the equilibrium Al–Ni diagram (Figure 7b) through points 1–4, and for inter-dendritic regions through points 5–8. The diagram shows that the temperature difference between points 2 and 3 is higher than between points 6 and 7. This means that in the inter-dendritic regions, the difference between the temperature of the solidus and solvus was lower (Figure 7a), which is related to the shorter time between the beginning of the creation of vacancies and the beginning of transformation *γ* → *γ* + *γ*′. It follows that the vacancies created and collected in the *γ* phase near LAB will only partially relax before the transformation *γ* → *γ* + *γ*′. This means that higher quantities of vacancies will be inherited by the *γ*′ phase near the LAB.

The dislocations density in area 2 and 3 (Figure 2d) can be calculated based on the values of misorientation angles of LABs and LAB-like defects. It can be assumed that for Ni-based superalloys, the LABs passed through the *γ* phase between *γ*′ precipitates in the area indicated, for example as *W* (Figure 7c) which had a value of about 2.5 μm. The assumption may be based on the fact that LABs are created in the *γ* phase, before the *γ*′ formation. The broken line of the LAB (Figure 7c) suggests the occurrence of both edge and screw dislocations in the LABs. The spacing of dislocations, separate for edge and screw type, can be expressed by Equations (5) and (6), respectively:(5)$$D_{e}=\left|\vec{b}\right|/\sin\varphi ,$$
(6)$$D_{s}=\left|\vec{b}\right|/2\sin\varphi ,$$
where *φ* is the LABs or LAB-like misorientation angle and *b* is the Burgers vector. The Burgers vector for *γ* phase is $\left|\vec{b}\right|=a\sqrt{2}/2$, where *a* is the lattice parameter. The dislocations density may be determined for macroscopic LABs and LAB-like defects. Taking into account the *W* value, the edge and screw dislocations density for macroscopic LABs may be expressed by Equations (7) and (8), respectively:(7)$$\rho_{me}=\frac{1}{D_{e}\cdot W}$$
(8)$$\rho_{ms}=\frac{1}{D_{s}\cdot W}$$

The edge dislocations density calculated for the LAB-I in area 2 of the PALS measurement (Figure 2d) was *ρ_me_*_2_ = 3 × 10^10^ × 1/cm^2^ and the screw dislocations density was *ρ_ms_*_2_ = 6 × 10^10^ × 1/cm^2^. In addition to the macroscopic LABs, there were micro-LABs located between misoriented neighboring dendrites [44]. Taking into account the TAS value, the edge and screw dislocations density for micro-LABs may be expressed by the relations, similar to that presented in Reference [44]:(9)$$\rho_{\mu e}=\frac{1}{D_{e}\cdot TAS}$$
(10)$$\rho_{\mu s}=\frac{1}{D_{s}\cdot TAS}$$

The edge dislocations density calculated for micro-LABs in area 2 was *ρ_μe_*_2_ = 2 × 10^7^ × 1/cm^2^ and the screw dislocations density was *ρ_μs_*_2_ = 4 × 10^7^ × 1/cm^2^. The edge dislocations density calculated for micro-LABs in area 3 was *ρ_μe_*_3_ = 1 × 10^7^ × 1/cm^2^ and the screw dislocations density was *ρ_μs_*_3_ = 2 × 10^7^ × 1/cm^2^. The abovementioned values are similar to the ones presented in Reference [44].

Except the macroscopic low-angle boundaries of the LAB-I and LAB-like type defects revealed by X-ray topography, in areas 1, 2, and 3 of the PALS measurements, there were micro-LABs located between neighboring misoriented dendrites. The micro-LABs were difficult to reveal using X-ray topography with conventional X-ray source, but such micro-LABs may be visualized and the misorientation angle may be determined using synchrotron radiation [44]. However, based on the PALS results, in area 1 with a lack of LAB-I-type and LAB-like defects, the presence of mono-vacancies suggest that they may be related to the abovementioned micro-LABs. Therefore, a comprehensive defect analysis was possible only with the use of different methods with different sensitivities to defects of various size scales.

## 4. Conclusions

Two types of macroscopic defects of misorientation character were formed outside the selector extension (SE) area. The first type of the LAB-like misorientation defects with misorientation angle of about 0.6° was formed by the lateral growth of some secondary dendrite arms near the surface of the selector–root connection. The second type were the LABs with higher misorientation angles of about 1° that were created in the periphery of the root transverse section. In the SE area of the root, the macroscopic defects of the low-angle boundaries (LABs) type were not formed. It was related to undisturbed axial growth of primary dendrite arms directly from the selector.It was found that in the SE area of the root, only Ni mono-vacancies occurred. There were two types of point defects in the root part, outside the SE area: Ni mono-vacancies with a predominant concentration, and the vacancy clusters. In addition, the local environment of the defects in atomic scale was different in these areas. The vacancy clusters occurred in the areas where macroscopic LABs and LAB-like defects were present as opposed to the SE area. It can be concluded that the vacancy clusters were related to LABs and LAB-like defects. Additionally, the areas with the LABs of higher misorientation angles than LAB-like defects differed in higher concentrations of the mono-vacancies and the vacancy clusters. It may be related to the increased dislocation density of the LABs.A combination of X-ray diffraction topography and positron annihilation lifetime spectroscopy allowed to determine the relation between the LAB-type defects and the vacancy-type defects created in the root areas of CMSX-6 single-crystalline as-cast blades.

## Figures and Tables

**Figure 1 materials-12-00870-f001:**
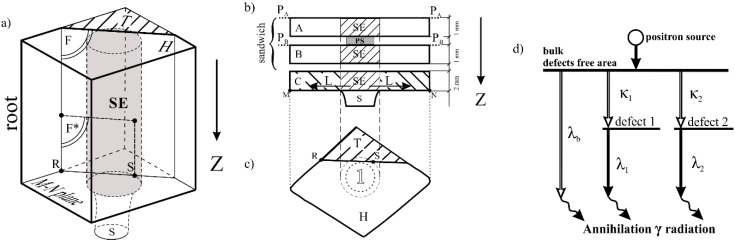
(**a**) Cut-off scheme of the root fragment T by plane *F* parallel to the withdrawal axis Z and the sections shape of the specimens of fragment H (**b**) parallel and (**c**) perpendicular to Z. P_A_–P_A_ and P_B_–P_B_—the surfaces from which the X-ray topograms were obtained, PS—the positron source for positron annihilation lifetime spectroscopy (PALS) measurements in area 1, S—selector, SE—selector extension area of the root; (**d**) scheme of the positron annihilation processes according to the three-state trapping model. Horizontal lines represent the delocalized state of positron in bulk material (defect-free area) and its two localized states on two types of defects. *λ_b_* is the annihilation rate of the delocalized (free) positron in the bulk, *λ*_1_ and *λ*_2_—the annihilation rates in two different types of defects. *κ*_1_ and *κ*_2_—the trapping rates in defects.

**Figure 2 materials-12-00870-f002:**
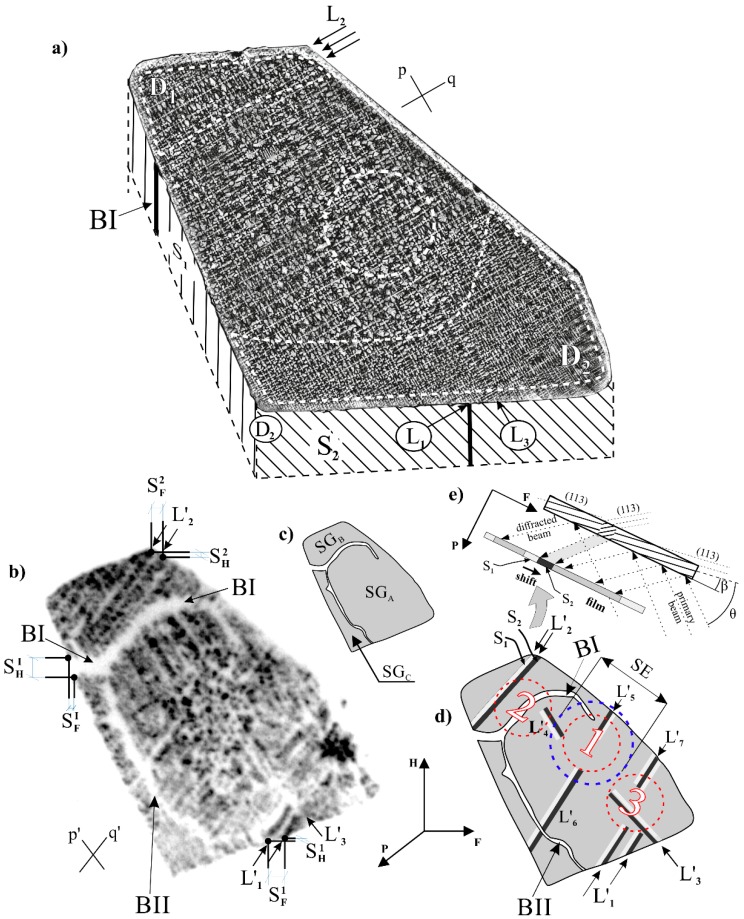
(**a**) Example of a macro-SEM image obtained from the P_A_–P_A_ surface; *p* and *q*—the directions of dendritic arms; (**b**) example of the topogram obtained from the P_A_–P_A_ surface; BI, BII—the contrast bands visualize the macroscopic low-angle boundaries LAB-I and LAB-II; *p*′ and *q*′—the directions of the contrast bands in the topogram; axis H and F lie in the plane of the microsection and the P axis is perpendicular to them; (**c**) scheme of sub-grains SG_A_, SG_B_, and SG_C_ visualized in the topogram; (**d**) scheme of the topogram with marked areas 1, 2, and 3 for PALS measurements; SE—selector extension area of root; (**e**) scheme of the contrast bands formation in the topogram.

**Figure 3 materials-12-00870-f003:**
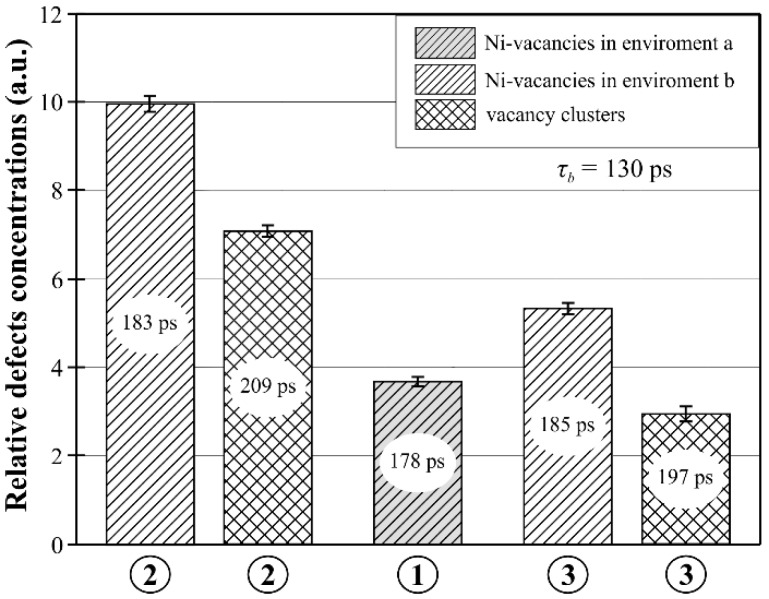
The relative concentration of different types of defects in areas 1, 2, and 3 with error bars and the values of positron lifetime.

**Figure 4 materials-12-00870-f004:**
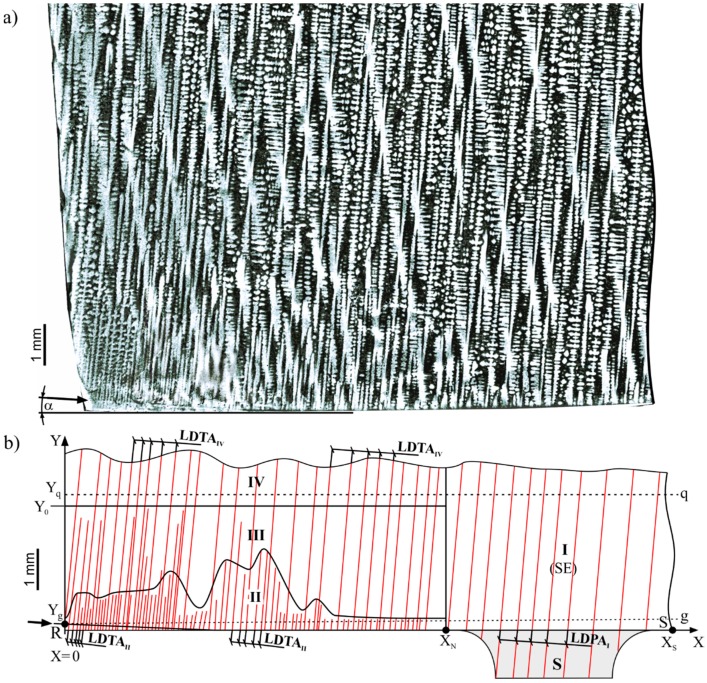
(**a**) The dendritic structure visualized on the surface *F** of the root part T; (**b**) the dendritic array scheme with growth areas marked as I, II, III, and IV.

**Figure 5 materials-12-00870-f005:**
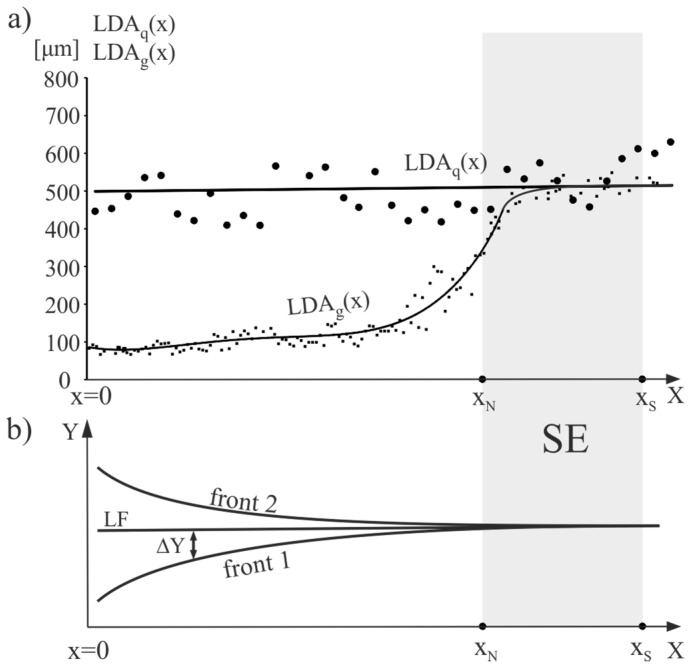
(**a**) The LDA_g_(x) and LDA_q_(x) relations; (**b**) hypothetical shape of crystallization fronts. LF—linear front perpendicular to the withdrawal direction.

**Figure 6 materials-12-00870-f006:**
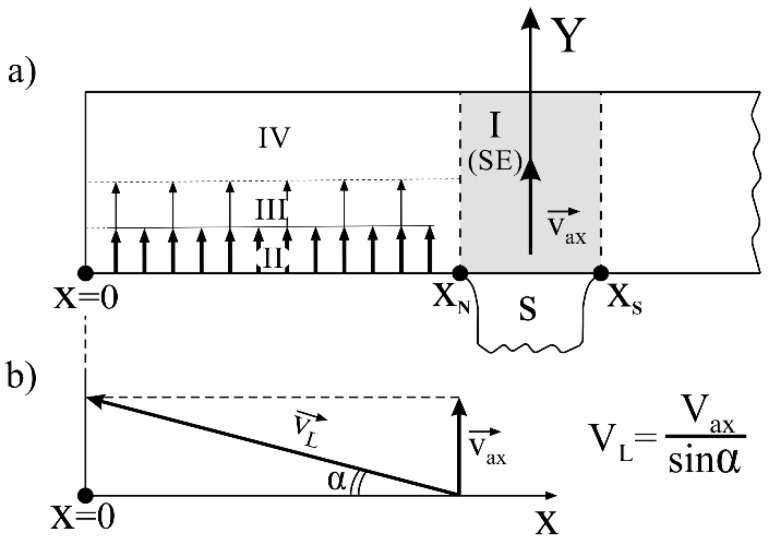
(**a**) Scheme of axial growth of dendrites in growth areas I–IV presented in Figure 4; (**b**) scheme that illustrates the relation between axial–*V_ax_* and lateral–*V_L_* rates of dendrite growth in area II. The *α* angle and thickness of growth area II are increased for figure clarity. Y—the direction of the crystallization front movement.

**Figure 7 materials-12-00870-f007:**
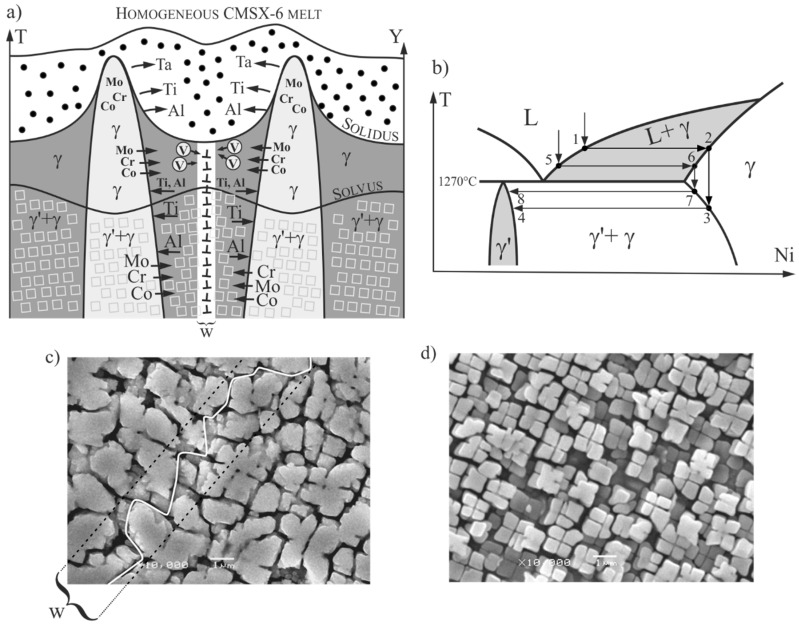
(**a**) Scheme of alloying elements’ distribution in the LAB area; (**b**) fragment of Ni–Al phase diagram; (**c**) example of *γ*/*γ*′ structure typical for areas 1, 2, and 3 of the PALS measurements in inter-dendritic regions and (**d**) in dendritic regions. Dotted area—diffusion area of liquid, W—the hypothetical LAB area, V—vacancies.

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
