# Peer review of "Defect Creation in the Root of Single-Crystalline Turbine Blades Made of Ni-Based Superalloy"

_materials, 2019, doi:10.3390/ma12060870_

Reviewer 1 Report

it is difficult to follow the results pesentation with different denomination of differents zones:

- the results presented in figure 3 and in figure 7 come from the same areas 1, 2 and 3 ? If yes, it is necessary to precise.

- the results presented in figure 4 and in figure 6 come from the same areas I, II and III ? If yes, it is necessary to precise.

- comparing to figure 1 and 2, it is extremely important to indicate, explicitely, the different zones 1,2,3 and I, II, III.

- it is also important to indicate the difference between zones 1/2/3 and I/II/III.

It is also important to cite some papers from Materials (ISSN 1996-1944) to justify the relevance of the treated subject concerning the journal.             

Author Response

Thank you very much for the valuable referee’s comments to our paper entitled: “The defects creation in root of single crystalline turbine blades made of Ni-base superalloy” - Manuscript Number:  materials-448300

The conclusions have been corrected according to the suggestions. The corrections in the discussion have been also made for clarifying the results and findings. Minor changes have been applied to the figures and captions, related to the different denomination of PALS measurement areas (1,2,3) and microstructure areas (I, II, III, IV) that indicate different zones.

In reference to the recommended major revision in the results presentation, the following changes with regard to denomination of different zones were made.

1. The results presented in figure 3 and in figure 7 come from the same areas 1, 2 and 3 of PALS measurements. It have been precise by changes in Fig. 7 caption and in the “Results and discussion” section.

2. The results presented in figure 4 and in figure 6 come from the same areas I, II, III and IV of dendrite growth and are related to the dendritic structure. It have been precise by changes in Fig. 4 and Fig. 6 captions and in the “Results and discussion” section.

3. In the Fig. 2, denotations of contrast bands visualizing low angle boundaries (I, II) were changed to BI and BII. Additionally, the denotation of D1 and D2 areas has been changed for improve the figure clarity.

Accordingly, several minor changes have been made in Figures 2, 4, 6 and 7.

According to the reviewer’s note, some additional references from journal Materials were added to introduction: [19] and [21].

Thank you very much for reviewer comments. I hope that revised manuscript will be accept and approve for publication.

Reviewer 2 Report

The authors made valuable contributions by carrying out an analysis of defect generation in the root of single crystalline turbine blades of Ni-base superalloy. The manuscript is recommended to be published as it is.

Author Response

Thank you very much for the valuable referee’s comments to our paper entitled: “The defects creation in root of single crystalline turbine blades made of Ni-base superalloy” - Manuscript Number:  materials-448300

Minor changes for correct spelling mistakes and for English improvement have been applied in the whole manuscript text.

Reviewer 3 Report

It is obvious that the authors presented valuable data acquired by X-ray diffraction topography and positron annihilation on single crystal Ni-based superalloy casting pieces. Therefore, the publication of the manuscript is recommended.

However, it should be also mentioned that the manuscript is extremely difficult to read and understand. The authors may want to improve their writing and presentaton so that the paper can be read by many readers of different background. Not only the way of presentation, English grammar should be carefully checked by a professional service provider.